# Quantifying Uncertainty in Food Security Modeling

**Syed Abu Shoaib** [1,*] **, Mohammad Zaved Kaiser Khan** [2] **, Nahid Sultana** [3] **and Taufique H. Mahmood** [4]

1   Department of Civil and Environmental Engineering, College of Engineering, King Faisal University, Al-Hofuf 31982, Al-Ahsa, Saudi Arabia

2   Water and Environmental Engineering, School of Computing, Engineering and Mathematics, Western Sydney University, Sydney, NSW 2751, Australia; zavedkaiser97@yahoo.com

3   School of Humanities and Languages, University of New South Wales, Sydney, NSW 2052, Australia; n.sultana@unsw.edu.au

4   Harold Hamm School of Geology and Geological Engineering, University of North Dakota, Grand Forks, ND 58202, USA; taufique.mahmood@und.edu

*   Correspondence: sabushoaib@kfu.edu.sa; Tel.: +966-056-343-0815

**Abstract:** Food security is considered as the most important global challenge. Therefore, identifying long-term drivers of food security and their connections is essential to steer policymakers determining policies for future food security and sustainable development. Given the complexity and uncertainty of multidimensional food security, quantifying the extent of uncertainty is vital. In this study, we investigated the uncertainty of a coupled hydrologic food security model to examine the impacts of climatic warming on food production (rice, cereal and wheat) in a mild temperature study site in China. In addition to varying temperature, our study also investigated the impacts of three $CO_2$ emission scenarios—the Representative Concentration Pathway, RCP 4.5, RCP 6.0, RCP 8.5—on food production. Our ultimate objective was to quantify the uncertainty in a coupled hydrologic food security model and report the sources and timing of uncertainty under a warming climate using a coupled hydrologic food security model tested against observed food production years. Our study shows an overall increasing trend in rice, cereal and wheat production under a warming climate. Crop yield data from China are used to demonstrate the extent of uncertainty in food security modeling. An innovative and systemic approach is developed to quantify the uncertainty in food security modeling. Crop yield variability with the rising trend of temperature also demonstrates a new insight in quantifying uncertainty in food security modeling.

**Keywords:** food security; modeling; uncertainty; metric; new insights; climate change

## 1. Introduction

Food security is an integral part of sustainable development. Many factors like climate change, COVID-19, lack of proper planning, aging farmers, massive bee die-off, soil erosion, genetic engineering, land development and uncertainty in future modeling make it a complex challenge. Understanding long-term drivers of food security and their interactions is needed to guide policymakers deciding on policies for future food security. Given the complexity and uncertainty of multidimensional food security, model-based scenario analysis is widely considered as a useful tool [1] for future projections and decision-making. A systemic approach is needed; instead of looking at individual indicators, their interlinkages must be considered. Efforts have also been made to improve the understanding of undernourishment drivers from a conceptual and an empirical perspective. Hasegawa et al. (2015) referred to the central drivers of future risk of food security and summarized that population growth and equality are essential elements in its long-term assessment [2]. However, to date, the empirical innovations have not been largely incorporated into quantitative perspective work. Scenario studies with a particular focus on food security frequently use a set of three indicators: food availability (kilocalories per person per day), hunger indicators (pervasiveness of undernourishment)

and food prices (mostly for cereals) [2–4]. While the Agricultural Model Intercomparison and Improvement Project (AgMIP) [5] has examined and narrowed the differences between models (in estimating long-term run impacts of climate change on agriculture: production, prices) through systematic model intercomparison [6–11], less progress has been made on the other dimensions (e.g., time-varying parameter) of food security.

Moreover, the key food security uncertainties identified by stakeholders are (i) equality, (ii) lifestyle and (iii) natural resources (sustainability), recognizing that the distribution of growth is crucial next to the sustainability of choices [11]. As the extent of uncertainty is wide and needs to quantify for the welfare of uncertain people living without food, this study developed an innovative approach that includes cross-correlation of crop yields with the future time domain, including different emission scenarios—the Representative Concentration Pathway, RCP 4.5, RCP 6.0, RCP 8.5—to quantify the uncertainty in food security modeling.

A first key contribution of this paper is to develop a methodological framework to quantify the spectrum of time-varying uncertainty in food security modeling, considering the variability in crop yield. Then, a second key contribution of this paper is accounting for the comprehensive dynamic character (variability in the future domain-changing correlations with different emission scenarios—the Representative Concentration Pathway, RCP 4.5, RCP 6.0, RCP 8.5) of the uncertainty in food security modeling.

This paper is organized as follows: Section 1.1 presents the uncertainty in the modeling process, and the following subsection describes the uncertainty in the food security model and the climate change impacts in different types of food security models used; Sections 2 and 3 describe the data and explain the methods of model simulation, and provide an evaluation of the proposed scheme, focusing on the extent of the uncertainty analyses; Section 4 discusses the results obtained with reference to crop yield with the selection of the different types of crops; and finally, Section 5 summarizes the key findings and outlines the scope of future work.

### 1.1. Uncertainty in Modeling Process

Uncertainty is an inherent feature of any modeling implementation, irrespective of the model requirements, context, features or agility [12]. This is particularly true in the modeling of food security systems, which are subject to a lack of perfect knowledge of economic processes and limited or imperfect observations. One of the preliminary steps in a modeling exercise is to identify the system of interest, subject to the model scope and limited knowledge of the system, to ultimately reduce uncertainty.

The sources of uncertainty are generally well-defined in modeling (Figure 1), even if they may not be easily quantified in any specific modeling task. First, there are errors in the measurements of meteorological or any climate-related forcing (mainly rainfall) due to many reasons [13–18], in particular, including the errors of interpolation from measurement points to the point of interest [17]. The errors of interpolation frequently come from sparse rainfall samples to capture the real rainfall variability and related interpolation model errors [19,20]. Measurement errors also exist in the discharge data [21] to which we train our model during calibration processes.

In fact, there are uncertainties in measurement/estimation of every component of a system balance to varying degrees, depending on the component [22–25], for instance, inadequate spatiotemporal resolution of snowpack data, river discharge, groundwater flow, land use change and human water abstraction/return flows. Considering catchment topography and vegetation patterns, the largest uncertainty in estimation could be within evapotranspiration, especially in the agroeconomic zone [26].

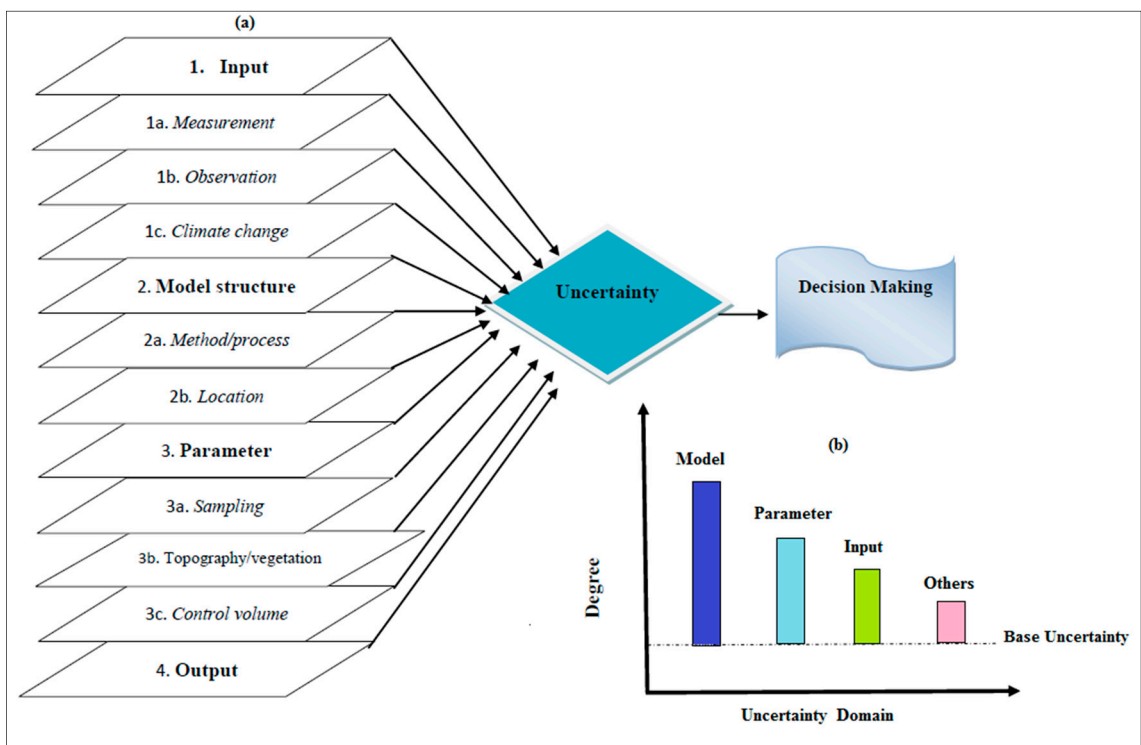

**Figure 1.** (**a**) Dimensions of uncertainty sources. (**b**) Probable degree of uncertainty in modeling.

Second, every component of a complex model is subject to uncertainties, even those that we usually would consider as accurate [27–32]. These structural uncertainties might differ depending on the type of model, distributed or lumped, size of subunits, etc. Uncertainty also varies depending on what models are used and where they are applied (e.g., in a natural, urbanized or rural catchment) [33]. In addition, the bases of all mathematical models are extracted from an equation, and frequently the equation solution will require simplifications, causing uncertainty. Oversimplification of the modeled system [16,34,35], or numerical error such as truncation error, might enhance with time-inducing, misleading results. Moreover, the numerical algorithm used for solving the equations could be sensitive to some conditions [36–38]. Simulating outputs from multiple model components and model choices can help to improve this.

We understand different uncertainty sources (Figure 1) but they are reliant on the identification of appropriate model hypotheses. The difficult problem of characterizing model structure adequacy is now attracting considerable attention, and recent work based on information theory may help to avoid those methods based on model structure selection [29].

The third source of uncertainty frequently cited is due to incorrect conceptualization and the values of parameters used [32]. In general, models incorporate many parameters that typically require calibration via field measurement or statistical methods [39]. The relative contribution of parameter uncertainty/structural identifiability to a model's total uncertainty can be significant, considering the type of model used and the catchment response [40]. Parameter uncertainty can result in a wide spectrum of values for any particular model output. However, significant research in model optimization has helped historically to reduce the impacts of parameter uncertainty. By retaining only realistic parameter values, the overall model uncertainty can be reduced [41].

Finally, output uncertainty is also considered as an important source of uncertainty. Usually, this source of uncertainty depends on the summation of input uncertainty, but does not correlate with prediction accuracy when used in a forecasting system [42]. Several output variables might be predicted in the modeling system aside from the water resources economic response, like water content in the soil profile or ground water level, which can

be used for retaining realistic parameter sets. Recognizing inherent uncertainties in these observations can help produce more realistic estimates of parameter uncertainty along with total model uncertainty [43].

Overall, uncertainty in food security modeling arises from several sources: model structure, parameters, observational input data and output used to drive and evaluate the model (Figure 1). A useful first step in identifying the sources of uncertainties is to evaluate how models are defined and constructed.

*1.2. Food Security Models and Climate Change Impacts*

1.2.1. Trends of Food Security Models

In the course of the past decade, simulation-based models (Table 1) of crop growth have increasingly been used to comprehend how climate change may affect the world's ability to produce food [44]. The International Food Policy Research Institute (IFPRI) has commenced a major sustained endeavor to analyze changes in the productivity of major crops across the entire world. The results are integrated into economic modeling efforts ranging from household to country-level economy-wide models to the global agricultural sector partial-equilibrium economic model known as IMPACT [45].

**Table 1.** Typical models used in the global food security setup and appraisal studies.

| Model | World Food System Model | Watersim | IMPACT | GLOBE | IMAGE | ABARES | Agrobiom |
|---|---|---|---|---|---|---|---|
| Type | CGE | PE | PE | CGE | IA | PE | Biomass |
| Economy coverage | Total economy | Agriculture and water | Agriculture | Total economy | Agriculture | Agriculture | Agriculture |
| Spatial scale | 34 Regions | 282 Sub-basins | 115 Regions | 19 Regions | 24 Regions, 0.51_0.51 grid | 37 Regions | 149 Regions |
| Sectoral scale | 10 Sectors | 32 Commodities | 32 Commodities | 12 Sectors | 12 Commodities | 33 Commodities | 5 Biomass categories |
| Institution | IIASA, Austria | IWMI-IFPRI, USA | IFPRI, USA | Oxford Brookes University, UK | PBL, The Netherlands | ABARES, Australia | INRA/CIRAD, France |
| Documentation | [44] | [46] | [45] | [47] | [48] | [49] | [50] |
| Food prices | Equilibrium prices | Equilibrium prices | Equilibrium prices | Equilibrium prices | n.a. | Computation and estimation of food security indicators. | |
| Calorie availability | No information on calculation | No information on calculation. | Post calculation using equilibrium food supply from model combined with calorie conversion factors | n.a. | n.a. | Equilibrium prices | n.a. |
| Undernourishment | Post estimation using the ratio of average national calorie availability, relative to aggregate national food requirements from FAO as inputs | n.a. | Post estimation using the ratio of average national calorie availability, relative to aggregate national food requirements from FAO as inputs | n.a. | Post calculation using calorie availability and FAO data on food intake. | n.a. | Post calculation using equilibrium food supply from model combined with calorie conversion factors |

Most interestingly, crop modeling launches at the field level, and scaling this up to the global level is challenging. As climate data need to be collated, processed and formatted, representative crop varieties and planting calendars must be chosen [46]. Global economic

models have been increasingly used to project food and agricultural developments for long-term time horizons, but food security aspects have often been limited to food availability projections [45]. In this paper, we propose a broader framework to explore the future of food security with a focus on crop yield and a reasonable proxy for quantifying the relation with temperature to crop yield for specific locations, considering available data.

The economy-wide computable-equilibrium model, MAGNET-IMAGE, captures economic and environmental interactions through the entire economic and environmental system, considered as high potential to predict [47]. On the other hand, the land-focused partial-equilibrium model, GLOBIOM, zooms in on the interlinkages between agricultural practices and the environment [48]. Both models (MAGNET-IMAGE and GLOBIOM) are (recursively) dynamic-, multi-country or region-based, and multiproduct models. A summary of the main characteristics of the models is presented in Table 1. The modeling framework is used to project developments in each future world between 2010 and 2050, in 10-year time steps. In this article, our analysis follows the 20-year time steps to demonstrate the uncertainty and correlations in food security modeling.

As there is a prolific supply of scenarios in the field of climate change [51], most food security studies use such climate-related scenario frameworks. For example, the Shared Socioeconomic Pathways (SSPs) distinguish five global pathways portraying the future evolution of key aspects of society defined along two axes: (i) socioeconomic challenges to mitigation and (ii) socioeconomic challenges to adaptation [52]. Some SSP studies have found that climate change and mitigation options have profound implications for food security [2,10,11]. These climate-focused SSPs are noticed to be deficient as enthusiastic food security scenarios in terms of addressing inequality by a diverse group of stakeholders. Saline-tolerant crops, drought-tolerant crops, water-saving agro-based farming and rotated crops can be good examples.

### 1.2.2. Climate Change Impact on Modeling Process

The unfolding impacts of climate change on hydrology and water resources, especially in the form of extreme hydrologic events (e.g., floods), highlight the need to determine the resilience of current approaches to hydrologic modeling and associated environmental and socioeconomic planning and management responses [53,54]. We have entered the era of the anthropocene, in which the cumulative impact of human activity has significantly affected the earth's ecosystems, environment and weather patterns. It is now evident that the nature of flood and drought extremes is changing as a result of global temperature rise [55,56].

Global mean atmospheric temperatures have risen one degree centigrade since preindustrial times and it is now generally acknowledged that such an increase in extreme weather is due to human-induced climate change. The recognition of climate change and its impacts has triggered discussions over potential compensation for those communities who have suffered losses and damage due to associated weather extremes [57]. Determining a valid loss and damage framework for climate change-induced impacts is, and will likely continue to be, a highly complex and contentious issue, particularly given the difficulties in definitively separating climate change-induced impacts from natural climatic variation. Climate change impacts include [58,59] slow onset events (such as salinity in coastal areas due to sea level rise), rapid onset events (such as cyclones and floods) and associated socioeconomic losses [60]. Such damage, in turn, causes forced displacement and migration, raising the increasingly urgent need to make financial provisions to support affected and displaced people around the world [58].

The assessment of climate change impacts on food production using water resources is made difficult by known uncertainties in the use of climate projections from Global Climate Models (GCMs) [61–63]. Currently, the general consensus among the scientific community is that overall annual precipitation may not change much, but subannual patterns of rainfall could change significantly, with greater precipitation in the wet season and less in the dry season [64–67]. Paradoxically, despite little change in annual rainfall

totals, this will likely lead to more extreme flooding and more droughts (and reductions in moderate floods) [68]. This scenario is already apparent, as rainfall patterns become more erratic and unpredictable around the world.

The three main sources of uncertainty in assessing climate change impacts using climate projections can be seen in GCM ensembles, i.e., uncertainties due to the selected model structure, emissions scenarios and natural variability. The Intergovernmental Panel on Climate Change (IPCC) gathers reviews and integrates GCMs in its Climate Model Intercomparison Project (CMIP). In the most recent fifth phase of the project, CMIP5 (released in 2013) advanced the spatial resolution and parameterization of its predecessor, CMIP3 (2010), but it still demonstrates significant uncertainty in climate projections at the global scale. A key question is how to integrate such uncertainty in the use of these projections for applications such as flood and food damage assessment processes at the local level, where floods occur. Recent work [69] has derived metrics for efficiently quantifying uncertainty in projections from CMIP5. The square root of error variance (SREV) specifies uncertainty as a function of time and space, and decomposes the total uncertainty in climate projections into its three sources. Such a methodology could prove to be a powerful metric for quantifying the extent of uncertainty in assessing climate change impacts at the catchment scale.

Uncertainty in food security modeling estimates can lead to significant over- or under-investment and can result in either needlessly expensive over-preparedness or dangerously inadequate flood mitigation and protection measures. As the uncertainties in estimates affect decision-making, quantifying this uncertainty can provide an insight into potential errors and help improve the decision-making process [70,71]. Food security assessment is an essential for resource management and supporting policy development.

Food security risks are expected to increase across the globe due to increasing socioeconomic development (leading to the growth of populations within climate risk zones and the impacts of subsidence and climate change) [72]. There has been speculation that the frequency and intensity of extreme events will increase with warmer temperatures. However, as any future projections are uncertain, does this new uncertainty alter food productions or yield damage estimates? There have been recent studies showing consideration of observational uncertainty in flood damage estimation [72–74] that integrate crop yield loss as part of food resources. Furthermore, there is evidence that nonconsideration of uncertainty in, say, rainfall, can result in a bias in the estimation of a derived variable such as streamflow [75,76], where a nonlinear transformation exists. This leads us to speculate that a similar bias is present in the case of food security modeling, considering uncertainty in the changing nature of climate, especially temperature. Analysis shows seasonal rainfall forecasts (Figure 2) with respect to rainfall simulated from the Predictive Ocean Atmosphere Model for Australia-(POAMA) sea surface temperature (SST) (POM-Rain) over the grid cells where at least one of the observed sea surface temperature anomalies (SSTA) indices has significant correlation at 95% confidence interval ($> |0.39|$) with the observed rainfall [77]. Projections of precipitation and temperature from GCMs are generally the basis for assessment of the impact of climate change, and seasonal precipitation forecasts help irrigators and water managers in planning and making decisions to maximize returns on investments and to ensure security of water as well as food.

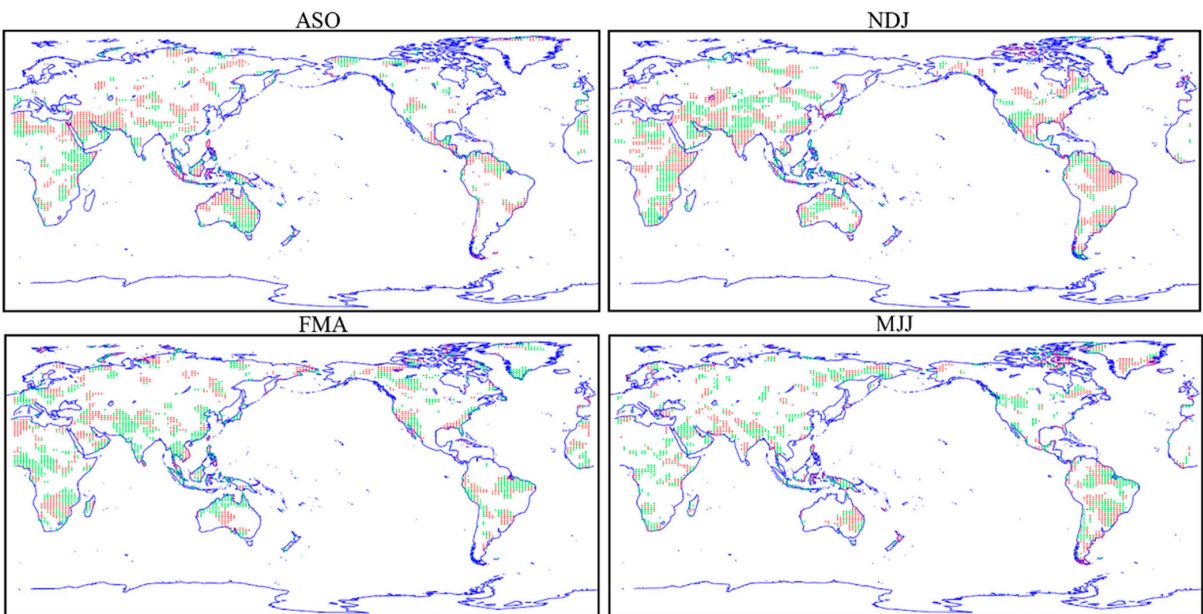

**Figure 2.** Rainfall simulated from multi-model sea surface temperature (SST)—seasonal rainfall forecasts with respect to rainfall simulated from POAMA SST (POM-Rain) over the grid cells where at least one of the observed SST anomaly (SSTA) indices has significant correlation at 95% confidence interval (> |0.39|) with the observed rainfall. Green- and red-shaded grid cells symbolize improvement and non-improvement by multi-model SST (MM-Rain), respectively [77]. The acronyms ASO, NDJ, FMA and MJJ represent the four seasonal variations. ASO—August, September, October; NDJ—November, December, January; FMA—February, March, April; MJJ—May, June, July.

## 2. Material and Methods—Quantifying Uncertainty in Food Security Modeling

### 2.1. Data and Study Area

In this study, the yearly crop yield data from China were taken to experiment with the analysis of uncertainty in food security modeling. The current study selected China as a study area due to readily available data and the region's vulnerability to current climate change scenarios. Crop yield data are available in the following link https://ourworldindata.org/crop-yields#yields-across-the-world. The historical or projected temperature data were sourced from the World Bank, https://climateknowledgeportal.worldbank.org/download-data. The emission scenarios RCP4.5, RCP6.0 and RCP8.5 were considered. There are four Representative Concentration Pathways (RCPs), which are RCP2.6, RCP4.5, RCP6.0 and RCP8.5. RCP2.6 has the lowest $CO_2$ emission scenario. Therefore, we excluded low emission scenario RCP2.6 to show the impact of a future warming climate on crop yield. Future climate impacts have focused on a warming scenario called "RCP8.5". This high-emissions scenario is frequently referred to as "business as usual", suggesting that is a likely outcome if society does not make concerted efforts to cut greenhouse gas emissions. For each of the emission scenarios, the projected data are available for 14 GCMs. The GCMs are bcc_csm1_1, ccsm4, cesm1_cam5, csiro_mk3_6_0, fio_esm, gfdl_cm3, gfdl_esm2m, giss_e2_h, giss_e2_r, ipsl_cm5a_mr, miroc_esm, miroc5, mri_cgcm3 and noresm1_m. The historical data period is 1961–2016 and the projected periods are 2020–2039, 2040–2059, 2060–2079, 2080–2099.

In addition to that, the daily rainfall and evapotranspiration data set was used in this study. The data set was derived from the Australian Water Availability Project (AWAP) [78,79], which is gridded to 0.050 × 0.050 and is extracted for the common 1980–2005 period. The accuracy of this data set is typically low where gage density is low, as is the case in central-west Australia, for instance [78]. The original meteorological data used in the AWAP product were supplied by the Bureau of Meteorology Australia (BoM). The AWAP platform uses model data fusion methods to combine both measurements and modeling. Daily rainfall data are available from 1900 to present, temperatures from

1911 to present and solar irradiance from 1990 to present. Gridded rainfall and potential evapotranspiration data used in this study are available from the following links:

(i)   for rainfall variability (http://www.bom.gov.au/web03/ncc/www/awap/rainfall/ totals/daily/grid/0.05/history/nat/\{startdate\}\{enddate\}.grid.Z'')

(ii)  for potential evapotranspiration variability (ftp://ftp.eoc.csiro.au/pub/awap/Australia_ historical/Run26j/FWPT/)

### 2.2. Models and Methods

To quantify the uncertainty in food security modeling, the following steps were considered.

Step 1: Selection of the conceptual hydrological model was based on catchment suitability. Four parent conceptual hydrological models (TOPMODEL, ARNOXVIC, PRMS, SACRAMENTO) were used in this experimentation. The dynamically dimensioned search (DDS) algorithm was used for parameter optimization process [80]. To estimate the food security modeling uncertainty, we analyzed the input uncertainty of the catchments using the Quantile Flow Deviation (QFD) metric [80,81]. Then the estimated input uncertainty was used to calculate model structure and parameter uncertainty.

Step 2: Obtained yearly crop yield (e.g., rice, cereal and wheat) for a country (e.g., China, Australia). The term cereal only represents a lumped term for a group of crops. Our data provider did not include details about the crop types in this group.

Step 3: Obtained historical monthly mean temperatures of a country; then converted to yearly mean temperature.

Step 4: Fit the linear regression between crop yield (Y) and yearly mean temperature (X).

Step 5: Or, Fit the nonlinear regression between crop yield (Y) and yearly mean temperature (X).

Step 6: For future scenarios, estimated crop yield from the projected yearly mean temperature. This process was to be done for 16 ensembles and three emission scenarios (Representative Concentration Pathway, RCP 4.5, RCP 6.0, RCP 8.5) to quantify the uncertainty in food security modeling),

Step 7: Generated the uncertainty estimation plot.

The modeling framework was used to project crop yields in future climate scenarios between 2020 and 2100, in every 20-year time step. In this statistical-based model, the predictand is yearly crop yield (e.g., rice, cereal and wheat) and the predictor is mean annual temperature. Then a linear regression was fit over the historical period (1961–2016). The spatial scale was over a whole country, e.g., China. The equation for linear regression was:

$$Y = mX + c \tag{1}$$

Here, Y is the predictand, crop yield; X is the predictor, temperature.

Using the historical period of data set, m (slope) and c (intersection) were determined. Then future temperature scenarios were used to estimate crop yield.

## 3. Results and Analysis

### Uncertainty Due to Model Structure

The median of projected yield for rice, wheat, cereal is likely to be increased in comparison to the historical period. As the years pass by, the GCMs suggest that the mean temperature over China is increasing. Increasing emission of carbon dioxide is likely to increase temperature, thus, our results show that the overall trend of crop production will continue to increase in a future warming climate (Figure 3A–C). This is conditioned on no reduction in land use for a particular crop, at least current similar quality of fertilizer and no impact of climate change on precipitation.

As part of the analysis, the correlation between historical crop yield and historical temperature is evaluated first (Figures 4–6). The value of correlation for different crops like wheat, rice and cereal shows a positive trend and that made the analysis of quantifying uncertainty in food security modeling worthy. The mean of historical yield for rice, wheat

and cereal is shown with the parallel dotted line. The value of the yield for each crop varies according to the land use and production capacity of the catchment.

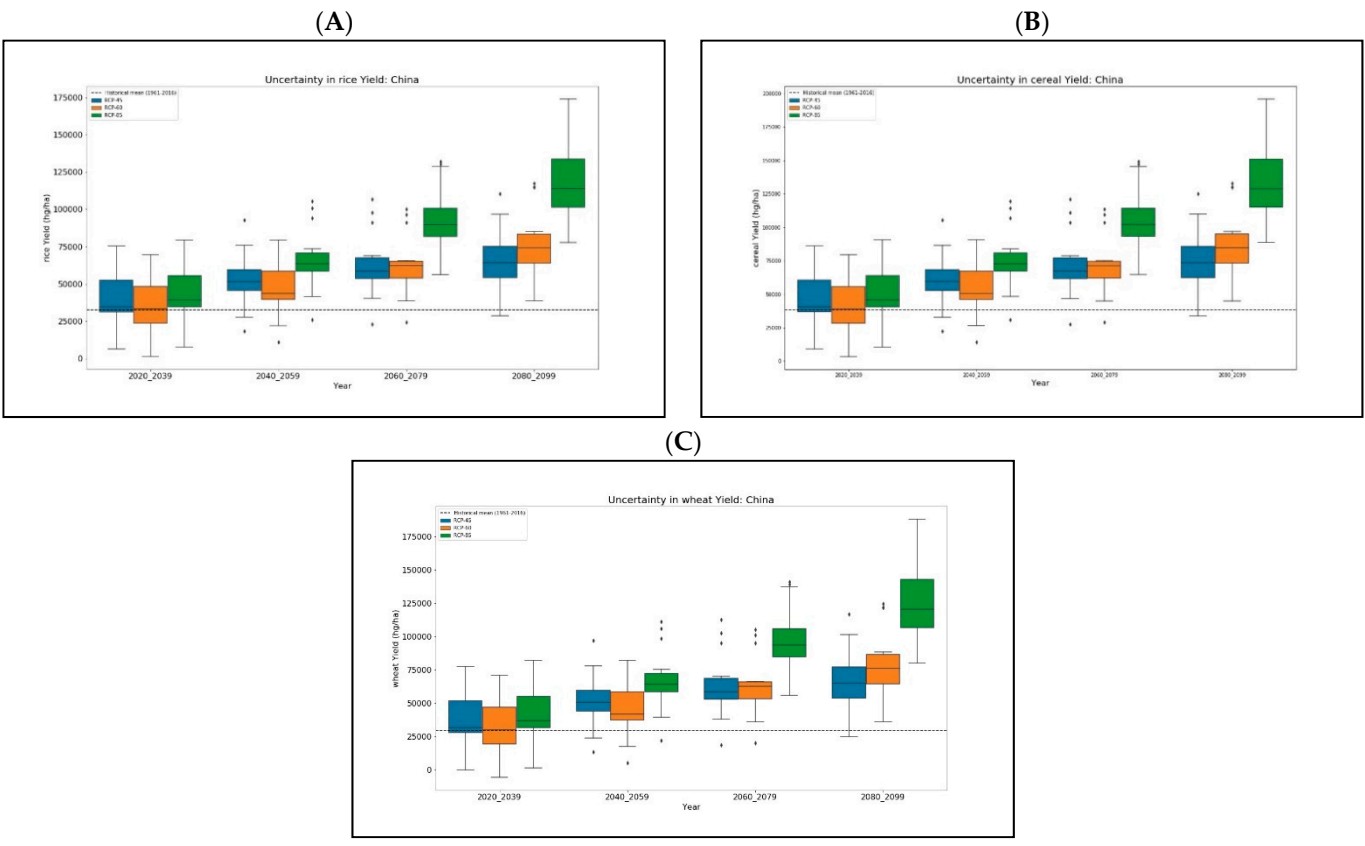

**Figure 3.** (**A**) Crop yield uncertainty projection in China—case Rice. (**B**) Crop yield uncertainty projection in China—case Cereal. (**C**) Crop yield uncertainty projection in China—case wheat.

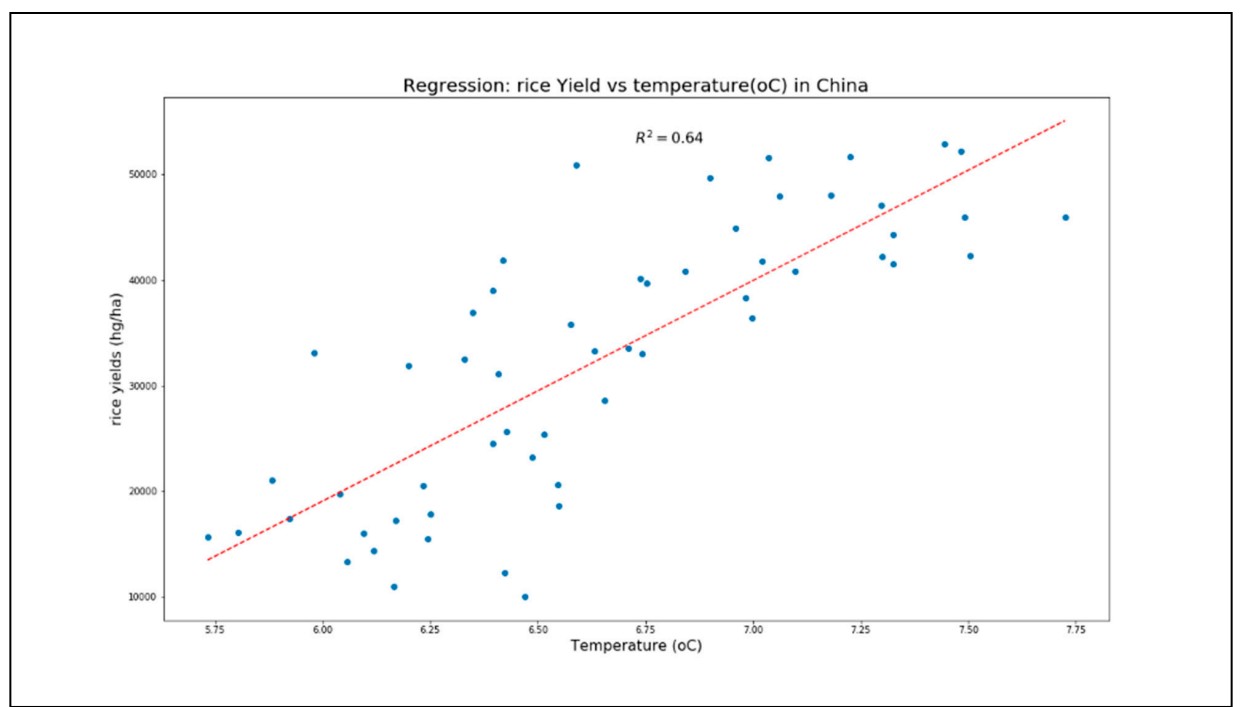

**Figure 4.** Crop yield—Rice vs. temperature (in °C) in China.

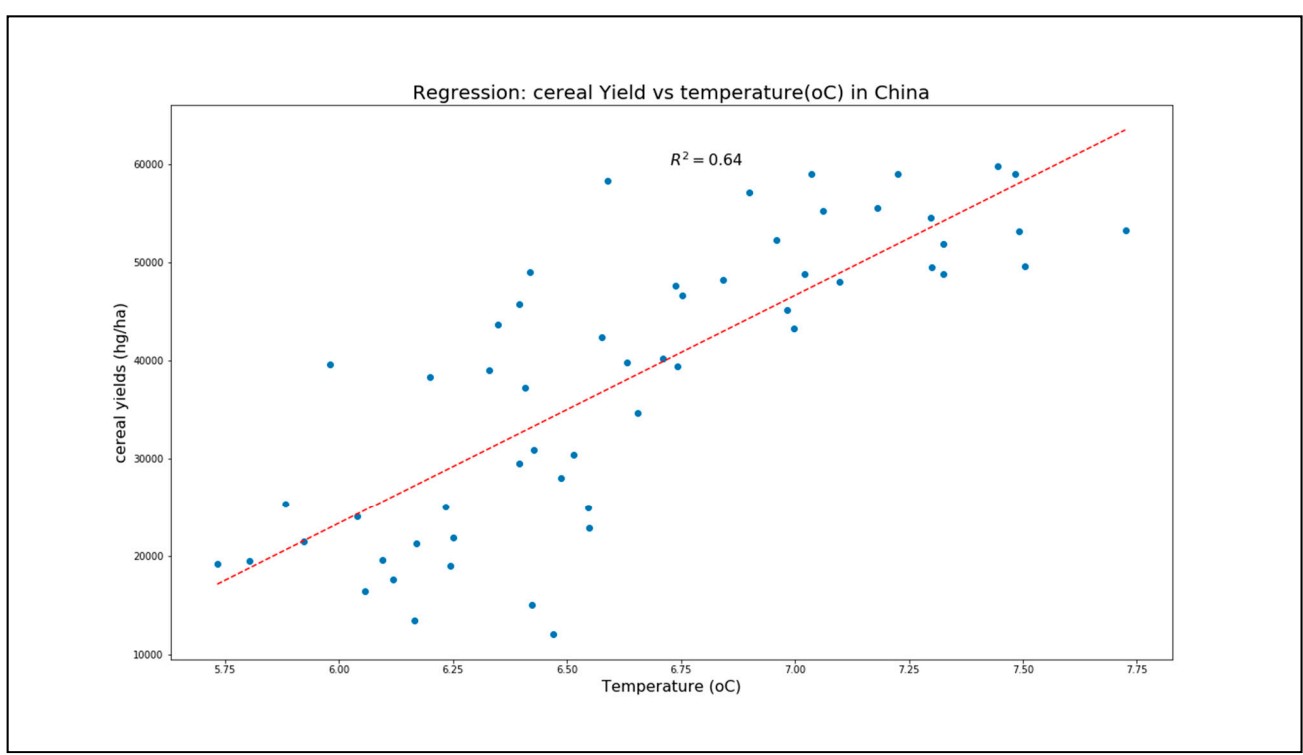

**Figure 5.** Crop yield—Cereal vs. temperature (in °C) in China.

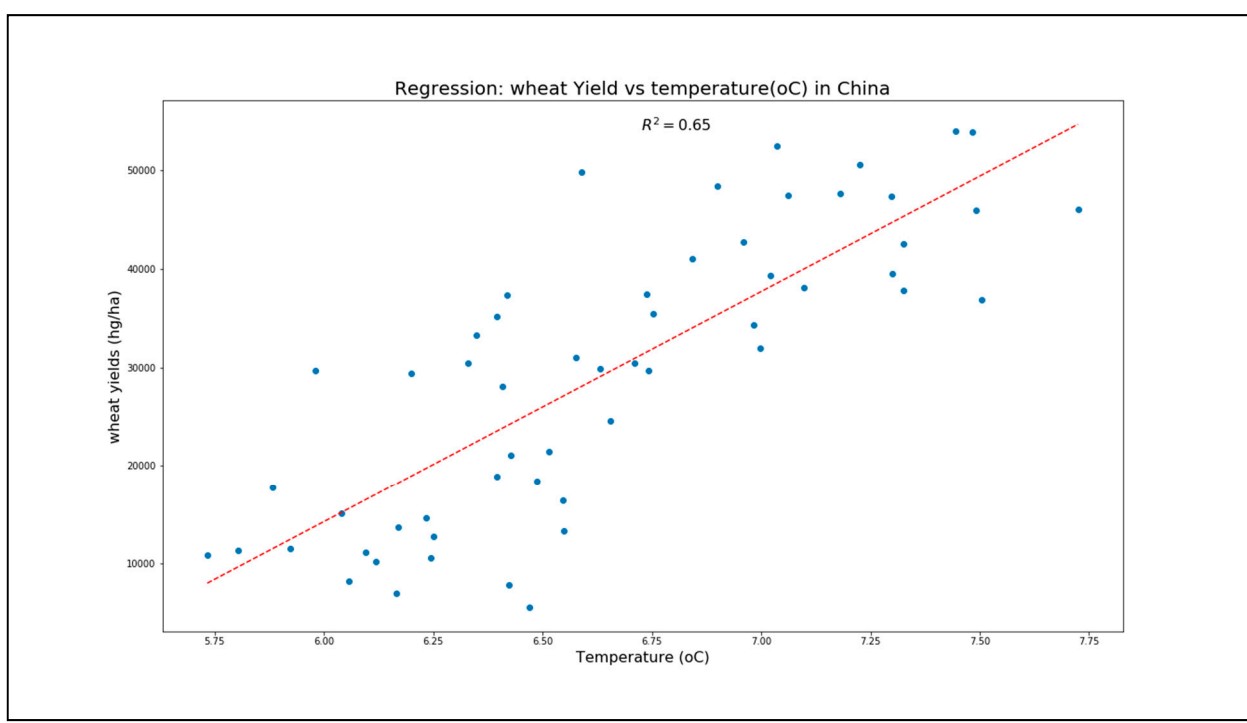

**Figure 6.** Crop yield—Wheat vs. temperature (in °C) in China.

## 4. Discussion—Uncertainty in Food Security Modeling

Emission scenarios for the Representative Concentration Pathways RCP 4.5, RCP 6.0, RCP 8.5 are analyzed as part of the experiment to visualize the extent of uncertainty in future food security. The results show significant uncertainty (Figure 3A–C) in the future yield of crops in China that needs to be addressed in the policy and planning process to combat any unusual situations.

Availability of data and resource management are vital in quantifying uncertainty in food security modeling. We have analyzed the input uncertainty of four different catchments: (i) Richmond catchment at New South Wales (NSW); (ii) Seventeen Mile catchment at Northern Territory; (iii) Buchan River catchment, at Melbourne, Victoria; (iv) Barambah River catchment at Queensland in Australia and it clearly shows that the availability of more gauge information reduces the uncertainty (Figure 7). Four parent conceptual hydrological models (TOPMODEL, ARNOXVIC, PRMS, SACRAMENTO) are used in this experimentation with the application of a dynamically dimensioned search optimization algorithm [80]. Analysis of input uncertainty of four different catchments in Australia shows the variability in uncertainty as a result of streamflow uncertainty. The Quantile Flow Deviation (QFD) metric [80,81] is used to estimate the input uncertainty as compared to model structure and parameter uncertainty.

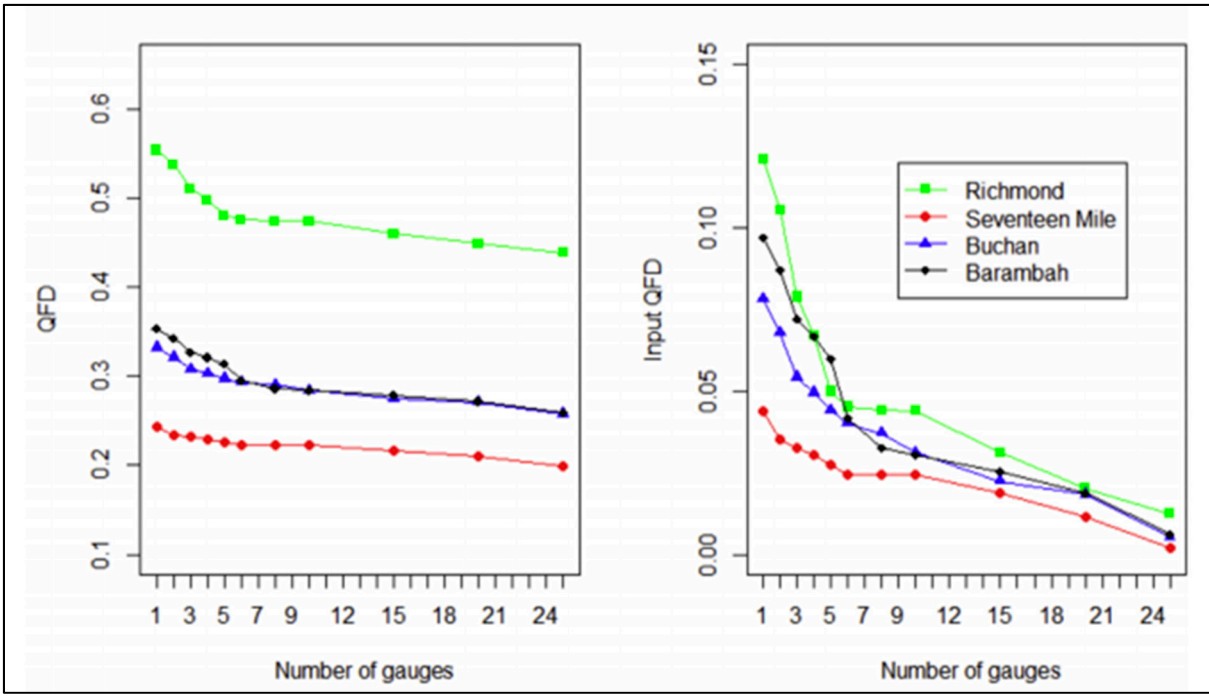

**Figure 7.** Experimental investigation shows that the augmented number of gauges reduce the overall uncertainty calculated through the Quantile Flow Deviation metric (QFD) [80,81]. Here, we showed how we can quantify uncertainty, considering the availability of data across four different catchments in Australia, named as (i) Richmond catchment at New South Wales (NSW); (ii) Seventeen Mile catchment at Northern Territory; (iii) Buchan River catchment, Melbourne, Victoria; (iv) Barambah River catchment at Queensland. The Australian Water Availability Project (AWAP) data set from the Bureau of Meteorology, Canberra, Australia is used for this analysis. Four parent hydrological models (TOPMODEL, ARNOXVIC, PRMS, SACRAMENTO) are used in this experimental setup.

On the other hand, in reality, achieving food security is much more complicated than having enough food available. Understanding future food security also requires insights into income distribution, purchasing power, political processes and institutional change [82,83]. Changes in global food security have been monitored using, for example, the Food and Agriculture Organization of the United Nations (FAO) suite of food security indicators [84]. Thus, the focus on food security models is based on realistic information.

In this paper, the future of global food security is assessed using a wider range of food security indicators like crop yield and temperature in a statistical multi-model framework. These indicators are typically used to evaluate current and ex-post trends in food security. However, we expand the coverage of these indicators in a consistent structural modeling framework. We complement this analysis with an assessment of the concomitant environmental impacts based on the Representative Concentration Pathways

RCP 4.5, RCP 6.0, RCP 8.5. Though possible land use change has significant implication in future food security modeling, considering drought and variability in soil moisture amount, we believe that much effort is needed for a reasonable conclusion, and our future research is focused on that particular aspect.

Our approach of using the increase in temperature as a predictor is based on past yield and temperature trends, and we extrapolate the temperature yield relationship. However, the intermediate processes like $CO_2$ fertilization and its effect on different types of crop productivity were not considered in the model. We believe the lack of representation of such processes in our model can be a source of additional uncertainty and should be considered in a future study. Temperature rise in climate scenarios is, of course, linked to rising $CO_2$ levels. However, the relationship between $CO_2$ and temperature dynamics can change between climate scenarios. The increasing temperature alone in a temperate climate like China's is not mainly the direct reason for increasing yields of C3 crops such as wheat, rather than rising $CO_2$ levels (beside production technology changes, including crop breeding effects and others). Rising temperatures alone would decline yields of cereals (C3 crops such as wheat) in temperate climates normally without genetic adaptation by new cultivars (shortening of growing cycles, less net assimilation rates), as shown by many crop modeling studies. Just maize (C4 crop) will rise yield mainly by increasing temperatures directly in temperate climates due to its much higher optimum growing temperature and already saturated $CO_2$ levels for maize. We believe future studies will consider and incorporate more biophysical processes with the options ingesting crop-to-crop variability.

## 5. Conclusions

In this study, we investigated the uncertainty of a coupled hydrologic food security model to examine the impacts of climatic warming on food production (rice, cereal and wheat) in a mild temperature study site in China. In addition to varying temperature, our study also investigated the impacts of three $CO_2$ emission scenarios—the Representative Concentration Pathway, RCP 4.5, RCP 6.0, RCP 8.5—on food production. Our ultimate objective was to quantify the uncertainty in a coupled hydrologic food security model and report the sources and timing of uncertainty under a warming climate using a coupled hydrologic food security model tested against observed food production. Our study shows an overall increasing trend in rice, cereal and wheat production under a warming climate. In particular, under the highest emission scenario (RCP = 8.5), a rapid increase in food production for all the crops is observed, and the maximum production is observed during the 2080–2099 period. The investigation of the intermediate processes and state variables reveals that the warming climate has minimal effects on water availability (soil moisture above wilting point) and substantial influences in accelerating biophysical processes and subsequently increasing crop productions. Further, we report the cascading effects of increasing temperature on parameter uncertainties and their influences on rice, wheat and cereal production. The current study also shows the relationship between crop production and mean annual temperature is generally linear, with other embedded complexities involved.

In the quantification of uncertainty in food security modeling, we provide an innovative flexible framework on the spectrum of uncertainty. Relative to the mentioned studies, our study is unique as it uses a combination of climate change uncertainty framework, a broad set expressing uncertainty through temperature, SSTA and rainfall, and we make a first step to include the uncertainty metric as the driver of outcomes. Future research will concentrate on following major areas: (i) variability in forcing data and its impact on food security modeling, (ii) understanding food security modeling impacts due to changing climate and the uncertainties embedded, and (iii) assessing changes in food security to hydro-ecologic systems based on consideration of the flow uncertainties.

In addition to that, time-varying parameter models for catchments with land use change have a huge implication in the uncertainty quantification process of food security modeling. The illustration is needed through an assessment of the possible trade-offs

between food and nutrition security and sustainability in each of the worlds. Our analysis method is flexible and can be disaggregated to examine any part of the modeling process, including the selection of certain model subroutines or certain forcing data. By considering multiple model structures, we are able to assess: (i) how the uncertainty varies across different case study catchments and (ii) how the uncertainty varies depending on the length of available observations.

**Author Contributions:** Conceptualization, S.A.S.; methodology, S.A.S., M.Z.K.K., N.S.; T.H.M.; software, S.A.S., M.Z.K.K., N.S.; validation, S.A.S., M.Z.K.K., N.S., T.H.M.; formal analysis, S.A.S., M.Z.K.K.; investigation, N.S.; resources, M.Z.K.K., N.S.; data curation, M.Z.K.K., S.A.S.; writing—original draft preparation, S.A.S., M.Z.K.K.; writing—review and editing, M.Z.K.K., N.S., T.H.M.; visualization, S.A.S., N.S.; supervision, S.A.S. and M.Z.K.K.; project administration, S.A.S.; funding acquisition, S.A.S. All authors have read and agreed to the published version of the manuscript.

**Funding:** This research is funded by the Deputyship for Research and Innovations, Ministry of Education, Saudi Arabia, Project number IFT20048.

**Institutional Review Board Statement:** Not applicable.

**Informed Consent Statement:** Not applicable.

**Data Availability Statement:** Data is available through the link provided in the Manuscript. All the data sources are widely used in the research community.

**Acknowledgments:** The authors extend their appreciation to the Deputyship for Research and Innovations, Ministry of Education, in Saudi Arabia, for funding this research work through project number IFT20048, with reference to the research grant number. The authors also acknowledge the Deanship of Scientific Research at King Faisal University for their kind assistance.

**Conflicts of Interest:** The authors declare no conflict of interest. The funders had no role in the design of the study; in the collection, analyses or interpretation of data; in the writing of the manuscript or in the decision to publish the results.

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
