# Peer review of "Quantifying Uncertainty in Food Security Modeling"

_agriculture, doi:10.3390/agriculture11010033_

Round 1

Reviewer 1 Report

Manuscript Title: Quantifying Uncertainty in Food Security Modeling

Manuscript Id: agriculture-1020102

Authors wrote the article nicely- Quantifying Uncertainty in Food Security Modeling. Food security is an integral part of many development agendas. Identifying long-term drivers of food security and their connection is essential for policy makers determining policies for future food security. Authors tired to explore the uncertainty of a coupled hydrologic-food security model to examine the impact of climatic warming on food production. They have clearly mentioned the models and methods. However, there are several things they have to improve.

  1. English editing is highly recommended
  2. 1 what is the reason behind Model has high uncertainty?
  3. Why did you choose china for the case study? Please explain in more detail.
  4. Page 8, Line 291-291, “This is conditioned on no reduction in land use for a particular crop, at least current similar quality of fertilizer and no impact of climate change in precipitation.”- Is it the valid assumption? I suspect on the validity of the results based on this assumption.
  5. I wonder why did you choose Australia for the hydrological models? I suggest you to conduct experiment in china for the hydrological models also to justify and unity of the research result.
  6. Article needs to be rewrite with redrawing the figures; the resolution of some of the figures is very low.

Author Response

Editor
MDPI-Agriculture

                                                                                                 22 December 2020

Dear Editor,

Subject: Submission of revised paper ‘Quantifying Uncertainty in Food Security Modelling’.

Thank you for your email dated 18th December 2020 enclosing the reviewers’ comments for the above-mentioned manuscript. We appreciate the valuable comments and suggestions on our manuscript from the Editor and anonymous reviewers.

We have carefully reviewed the comments and have revised the manuscript accordingly. Our responses are organised point-by-point for the comments from the reviewers which are provided in blue text. We attach here a tracked-changes version of our manuscript highlighting changes made directly to the manuscript.

 We believe that these revisions have substantially strengthened the manuscript and we hope that you find that the revised version is now suitable for publication. We look forward to hearing from you in due course.

Sincerely,

sabushoaib

Syed Abu Shoaib
Assistant Professor
Food Security, Water and Environment wing

Department of Civil and Environmental Engineering
King Faisal University, Saudi Arabia

Attached file:

  1. Response to reviewers
  2. Manuscripts with track changes

Response to reviewers comments – MDPI_Agriculture_1020102

We thank the Reviewers, Editor and the Assistant Editor for the comments which improve the readability and accuracy of this manuscript. We appreciate the constructive feedback and the thorough nature of the review that has gone into improving this manuscript. Below are our responses to each of the comments and how they have been dealt with.

A.
Response to Reviewer 1 Comments

 Authors wrote the article nicely- Quantifying Uncertainty in Food Security Modeling. Food security is an integral part of many development agendas. Identifying long-term drivers of food security and their connection is essential for policy makers determining policies for future food security. Authors tried to explore the uncertainty of a coupled hydrologic-food security model to examine the impact of climatic warming on food production. They have clearly mentioned the models and methods.

Thank you for your valuable comments. We gave significant attention to Quantifying Uncertainty in Food Security Modelling with innovative idea and addressed the points raised to make it worthy of publication in MDPI- Agriculture. We want to especially thank the reviewer for the detailed comments following that have helped us identify specific problems. Considerable effort has gone into improving the readability and clarity of the manuscript, and we believe the study is significantly improved as a result. 

  1. English editing is highly recommended.

We appreciate the reviewer’s comment and made sure we gave our utmost attention in proof-reading of the text. We gave significant attention to proof-reading of the text and addressed the points raised to make it worthy of publication.

  1. What is the reason behind Model has high uncertainty?

The collection of possible hydrologic models, input data, and parameter sets represent the variability in the modelling process (which might arise due to various choices the modeller has to make on which model to select, or might arise due to uncertainty in the parameter estimates). Our overall goal is to determine which of these uncertain choices has the biggest impact on simulated streamflow. Changes in the frequency of extreme events and uncertainty in their prediction increase concerns about the state of hydrological modelling [Clark et al., 2011; Gong et al., 2013; Nearing and Gupta, 2015; Westerberg et al., 2017]. Selection of model structure is very uncertain. Suitability model structure also varies from catchments to catchments. Land use change and extreme climatic events also contribute to the increase of high uncertainty.

  1. Why did you choose china for the case study? Please explain in more detail.

Availability of data and possible reference or sources made us choose china for the case study. To address this comment we have added following in line 237.

“The current study selected China as a study area due to readily available data and the region’s vulnerability to current and climate change scenarios.”

  1. Page 8, Line 291-291, “This is conditioned on no reduction in land use for a particular crop, at least current similar quality of fertilizer and no impact of climate change in precipitation.”- Is it the valid assumption? I suspect on the validity of the results based on this assumption.

In future, the agricultural land use can be changed, though the quality of fertilizer is expected to be improved with the advancement of technology. Therefore, less land use will be required to have more crop yield. However, the quantitative volume for these two factors under future climate scenario is quite difficult to predict. To avoid complexity, we ignored these two factors in our analysis. We agree that the precipitation scenario may be changed. It can be noted here that the precipitation even over the historical period (1961 - 2016) in a country, e.g., China does not show a clear signal whether the precipitation has increasing/decreasing trend. The relationship between the climate change and precipitation is non-linear. Hence, keeping our analysis simple, we intend to convey our message to the scientific community how climate change impact on temperature only can influence various crop yields in future.

  1. I wonder why did you choose Australia for the hydrological models? I suggest you conduct experiment in china for the hydrological models also to justify and unity of the research result.

We have applied the regionalization approach in this regard. We have selected four different catchments of Australia with diverse catchment properties as well as climatic and seasonal variability. Here we showed how we can quantify uncertainty considering the availability of data across four different catchment in Australia name as (i) Richmond catchment at NSW, (ii) Seventeen Mile Catchment at Northern Territory (iii) Buchan River Catchment, Victoria, Melbourne; (iv)Barambah river catchment at Queensland. AWAP data set from Bureau of Meteorology, Canberra, Australia is used for this analysis. Four parent hydrological models (TOPMODEL, ARNOXVIC, PRMS, SACRAMENTO) are used in this experimental setup. Again, it’s the availability of the complete sets of data considering climate variability. Use of regionalization approach, selecting diverse catchment of Australia and thereby application of four parent hydrological models (TOPMODEL, ARNOXVIC, PRMS, SACRAMENTO) are the representation of adopting the idea that we have presented in the study.

  1. Article needs to be rewrite with redrawing the figures; the resolution of some of the figures is very low.

Thank you for your comments. We have rewritten part of the manuscripts and incorporated high resolution figures (Figure 2 has been replaced) to make it worthy for publications.

Reviewer 2 Report

Dear author,

The paper was interesting to read as well as this is an interesting study on the food security field. I found it thoughtful, informative and reflective. Lots of broad considerations for the future. The team did well to produce so much content.

I attach a copy of the manuscript with some edits and comments or queries for more information or clarification.

I would recommend some clarification in some sections, such as introduction, food and climate change impacts, etc. In addition, few grammar mistakes should be corrected and references should be adapted to the journal's guidelines.

Below a list of suggestions to get the manuscript improved:

Line 54: Would be useful to present more information about the RCP emission scenarios, p.e. explaining what the 3 different RCP's mean, why these RCPs were chosen, why not others etc.

Line 150: The most studies presented in Table 1 use 10-year time steps and you opted for 20-year time steps, even though this choice is not clear in your justification.

Line 157: Would be interesting to add an example or two of such implications for food security.

Line 166: Punctuation (full stop) is missing at the end of this sentence.

Line 202: In order to make the figure clearer, it would be interesting to explain the acronyms ASO, NDJ, FMA, MJJ. Also, it would be more appropriate if this figure would be close to the place where it is first mentioned (line 223 or 230).

Line 233: Would it be appropriate to use more countries in order to make a more accurate analysis of the modelling explored?

Line 237: Could you explain why these three scenarios were chosen?

Line 241: Could you explain why you haven't added data from 2017-2019?

Line 242: Projected periods are precisely 19-year time steps. In line 150, it might be more appropriate to state this information rather than 20-year time steps?

Line 245: Would it be any constrains for this analysis to choose a daily rainfall and evapotranspiration data set period different from the historical period?

Line 290: Please check this sentence grammatically.

Line 312: This sentence is a bit confusing, it might be good to rephrase it. Here it's a suggestion: "....China that needs to be addressed....".

Line 317: This sentence is not clear. Punctuation should be checked.

Line 320: It should be "search"?

Line 324: Full stop missing.

Line 329: Grammar should be checked.

Line 342 and 347: Please, correct spelling.

Line 399:Please check guidance for references, they should follow the below example, for articles:

1. Author 1, A.B.; Author 2, C.D. Title of the article. Abbreviated Journal Name YearVolume, page range.

Author Response

  1. Response to Reviewer 2 Comments

The paper was interesting to read as well as this is an interesting study on the food security field. I found it thoughtful, informative and reflective. Lots of broad considerations for the future. The team did well to produce so much content.

I attach a copy of the manuscript with some edits and comments or queries for more information or clarification. I would recommend some clarification in some sections, such as introduction, food, and climate change impacts, etc. In addition, few grammar mistakes should be corrected, and references should be adapted to the journal's guidelines.

We appreciate your comments and agree with the suggestion. We have updated the Introduction as per your suggestions and believe it conveys the intent of the study in a much more transparent manner.

  1. Line 54: Would be useful to present more information about the RCP emission scenarios, i.e. explaining what the 3 different RCP's mean, why these RCPs were chosen, why not others etc.

We have included more information about the RCP emission scenarios, p.e. explaining what the 3 different RCP's mean, why these RCPs were chosen, why not others etc. Please see the updated line in the manuscripts [Line 244-250].

The Representative Concentration Pathways (RCPs)-Future climate impacts have focused on a warming scenario called “RCP8.5”. This high-emissions scenario is frequently referred to as “business as usual”, suggesting that is a likely outcome if society does not make concerted efforts to cut greenhouse gas emissions. Projecting future climate change involves assessing a number of different uncertainties. Some of these relate to the climate system, such as how sensitive the climate might be to increased concentrations of greenhouse gas in the atmosphere. Others involve the quantity of gases emitted, using energy system models to simulate different scenarios of future emissions.

There are four RCPs which are RCP2.6, RCP4.5, RCP6.0, RCP8.5. RCP2.6 has the lowest CO2 emission scenario. Therefore, we excluded low emission scenario RCP2.6 to show the impact of future warming climate on crop yield.

  1. Line 150: The most studies presented in Table 1 use 10-year time steps and you opted for 20-year time steps, even though this choice is not clear in your justification.

The climate change projection data used this analysis has 20 years timestep.  Even though some studies presented in Table `1 use 10-year time, we have used the 20 years time step to quantify the extent of uncertainty.in the modelling process. Data availability based on case study as well as model selection is the key of any experimental design.

  1. Line 157: Would be interesting to add an example or two of such implications for food security.

 Saline tolerant crops, drought tolerant crops, water saving agro-based farming, and rotate crops can be good examples for food security.

  1. Line 166: Punctuation (full stop) is missing at the end of this sentence.

Punctuation (full stop) is added at the end of the sentences. Thank you.

  1. Line 202: In order to make the figure clearer, it would be interesting to explain the acronyms ASO, NDJ, FMA, MJJ. Also, it would be more appropriate if this figure would be close to the place where it is first mentioned (line 223 or 230).

Thank you for pointing this important fact. The acronyms ASO, NDJ, FMA, MJJ represent the four seasonal variation. ASO- August, September, October; NDJ- November, December, January; FMA- February, March, April; MJJ- May, June, July. We agree to place the figure where you have mentioned.

  1. Line 233: Would it be appropriate to use more countries in order to make a more accurate analysis of the modelling explored?

Our food security project’s next phase has the intention to include more countries considering the availability of data and more institutional collaboration will be developed for a more accurate analysis of the modelling explored.

  1. Line 237: Could you explain why these three scenarios were chosen?

There are four RCPs which are RCP2.6, RCP4.5, RCP6.0, RCP8.5. From the figure below, we can see that RCP2.6 has the lowest CO2 emission scenario. Therefore, we excluded low emission scenario RCP2.6 to show the impact of future warming climate on crop yield.

Figure: Emissions of COâ‚‚ across the RCPs (left), and trends in concentrations of carbon dioxide (right). Grey area indicates the 98th and 90th percentiles (light/dark grey) of the values from the literature). The dotted lines indicate four of the SRES marker scenarios. SOURCE: van Vuuren et. al. (2011)

  1. Line 241: Could you explain why you haven't added data from 2017-2019?

We sourced the historical or projected temperature data from World Bank, https://climateknowledgeportal.worldbank.org/download-data. The available data for the historical period is 1961 – 2017. Therefore, we have not included data for 2017-2019.

  1. Line 242: Projected periods are precisely 19-year time steps. In line 150, it might be more appropriate to state this information rather than 20-year time steps?

The projection data is available for 2020-2039, 2040-2059, 2060-2079, 2080-2099. For example, over the period 2020-2039 implies data inclusive 2020 which means the data length has 20 timesteps.

  1. Line 245: Would it be any constrains for this analysis to choose a daily rainfall and evapotranspiration data set period different from the historical period?

We don’t see any constrains as we use the regionalization approach considering with similar catchment properties as well as climate data. That again remind us the importance of quantification of uncertainty in food security modelling.

  1. Line 290: Please check this sentence grammatically.

We have checked the Line 290 grammatically and updated accordingly.

  1. Line 312: This sentence is a bit confusing, it might be good to rephrase it. Here it's a suggestion: "....China that needs to be addressed....".

Thank you for your suggestion. We have modified accordingly.

  1. Line 317: This sentence is not clear. Punctuation should be checked.

Punctuation is corrected and the sentence is structured to make the sentence clear. Thank you.

  1. Line 320: It should be "search"?

Thank you, Yes, it should be search. We have updated it.

  1. Line 324: Full stop missing. Line 329: Grammar should be checked.

We appreciate your review. Full stop is placed in Line 324. Grammar is checked and updated in Line 329.

  1. Line 342 and 347: Please, correct spelling.

Thank you. We have corrected the spelling.

  1. Line 399: Please check guidance for references, they should follow the below example, for articles:
  2. Author 1, A.B.; Author 2, C.D. Title of the article. Abbreviated Journal Name Year, Volume, page range.

We have updated the references according to the guidelines provided. Please see the reference section.

This manuscript is a resubmission of an earlier submission. The following is a list of the peer review reports and author responses from that submission.